# Avascular Necrosis of the Talus: Diagnosis, Treatment, and Modern Reconstructive Options

**DOI:** 10.3390/medicina60101692

**Published:** 2024-10-15

**Authors:** Michał Jan Kubisa, Marta Gabriela Kubisa, Karol Pałka, Jakub Sobczyk, Filip Bubieńczyk, Paweł Łęgosz

**Affiliations:** 1Orthopedic and Traumatology Department, Medical University of Warsaw, 02-091 Warszawa, Poland; kubisa.michal@gmail.com (M.J.K.);; 2Orthopedic and Sport Traumatology Department, Carolina Medical Center, 02-757 Warszawa, Poland; jakub.sobczyk@carolina.pl; 3Medical University of Silesia, 40-055 Katowice, Poland; karol.palka@gmail.com; 4Orthopedic and Traumatology Department, Pomeranian Medical University, 70-204 Szczecin, Poland

**Keywords:** talar avascular necrosis, total talar replacement, total ankle replacement, ankle alloplasty

## Abstract

Talar avascular necrosis (AVN) is a devastating condition that frequently follows type III and IV talar neck fractures. As 60% of the talus is covered by hyaline cartilage, its vascular supply is limited and prone to trauma, which may eventually lead to AVN development. Early detection of AVN (Hawkins sign, MRI) is crucial, as it may prevent the development of the irreversible stages III and IV of AVN. Alertness is advised regarding non-obvious conditions that may cause this complication (sub chondroplasty, systemic lupus erythematosus, diabetes mellitus). Although, in stages I–II, AVN may be treated with non-surgical procedures (ESWT therapy, non-weight bearing) or joint-sparing techniques (core drilling, bone marrow aspirate injections), stages III–IV require more advanced procedures, such as joint-sacrificing procedures (hindfoot arthrodesis/ankle arthrodesis), or replacement surgery, including total talar replacement (TTR) or combined total ankle replacement (TAR). The advancement of 3D-printing technology and increased access to implant manufacturing are contributing to a rise in the production rates of third-generation total talar prostheses. As a result, there is a growing frequency of alloplasty procedures and combined total ankle replacement (TAR) surgeries. By performing TTR as opposed to deses, the operator avoids (i) delayed union, (ii) a shortening of the limb, (iii) a lack of mobility, and (iv) the stiffening of adjacent joints, which are the main disadvantages of joint-sacrificing procedures. Simultaneously, TTR and combined TAR offer (i) a brief period of weight-bearing restriction, (ii) quick pain relief, and (iii) preservation of the length of the limb. Here, we summarize the most up-to-date knowledge regarding AVN diagnosis and treatment, with a special focus on the role of TTR.

## 1. Introduction

Talar avascular necrosis (AVN) is a challenging condition in which, despite the most meticulous treatment, the outcome and patients’ overall well-being measured using the AOFAS scale remain variable. As in other avascular necroses, its cornerstone comprises the impairment of the blood supply to the talar bone. In most cases, it occurs as a complication of a talar neck fracture; however, its etiology also encompasses a wide range of systemic diseases, such as systemic lupus erythematosus (SLE), type 2 diabetes (DM II), and steroid therapy. With its post-traumatic and atraumatic etiology, as well as its many advancement stages, the approach to treatment should be personalized. In recent years, many variations of surgical treatments for talar AVN have been developed, including total talar replacement (TTR) and combined total ankle replacement (TAR). The aim of this review is to summarize and present causes, common presentation, and treatment options, with a focus on total talar and total ankle alloplasty in the treatment of talar avascular necrosis.

## 2. Etiology

Due to multiple articulations with the tibia, fibula, calcaneus, and navicular bones and little attachment of ligaments and tendons, 60% of the talus is covered by hyaline cartilage [1]. The limited non-articular surface contributes to a poor penetration area for the blood vessels, such as the dorsalis pedis artery, the posterior tibial artery (through branches in the sinus tarsi), and the perforating peroneal artery, as presented in Figure 1 and Figure 2 [2].

Without oxygen and nutrients, osteocytes undergo apoptosis, which presents as necrosis of the bone and can lead to the collapse of the talar dome and degeneration of the other parts [3]. As intraosseous anastomoses are variable among individuals, some patients are more prone to suffering from talus AVN [4]. AVN develops due to a complete lack of or limited blood supply to the bone, which is secondary to (i) traumatic or (ii) atraumatic loss of blood supply.

Talus fractures account for 0.1% to 2.5% of all fractures and 3% to 5% of foot and ankle fractures [5]. Traumatic AVN develops as a consequence of talar fractures. Two scales are used for prognoses of AVN occurrence: Hawkins classification (in Canale and Kelly’s modification) (talar neck fractures) and Marti–Weber classification (all talar fractures) [6]. The classifications are presented in Table 1.

Talar fractures occur mostly as a result of abrupt axial load and foot dorsiflexion during (i) motor vehicle collisions (MVCs), (ii) motorcycle collisions, (iii) falls from height, (iv) pedestrians being struck by automobiles, (v) crush injuries, and (vi) athletic injuries [7].

The risk factors that are associated with an increased chance of developing an AVN post talar fracture include fracture type, tobacco smoking, open fractures, dual approach surgery, increased age, and increased BMI [8]. The subtalar dislocation in each case drastically multiplies the risk of AVN due to blood vessel impairment.

Cases of atraumatic AVN of the talus have been reported in patients treated chronically with corticosteroids, those suffering from alcoholism, systemic lupus erythematosus (SLE), sickle cell anemia, or hyperlipidemia, and those who have undergone renal transplants or irradiation.

Iatrogenic AVN is a relatively new concern that arises following subchondroplasty of the talar bone and calcium phosphate injections used in the treatment of OCD. These interventions can inadvertently impair blood supply to the talus, potentially leading to the development of AVN. It is crucial for clinicians to be aware of this risk when considering treatment options for patients with talar injuries or conditions [9,10,11].

The common denominator of all the above conditions is hindered blood supply to the talus [12].

To understand the progress in AVN treatments and apply proper treatment, one must understand each stage of talar AVN. In stage I of the Ficat and Arlet scale, the lack of circulation in the subchondral bone causes osteocyte necrosis due to anoxia 3 h post cessation of oxygen supply [13]. In the proliferative region of the necrotic zone, the blood vessels start to penetrate, which leads to the resorption of the dead bone (Hawkins sign) and the production of new living bone between the necrotic tissue [14]. In stage II, characterized by subchondral cysts and sclerosis, where subchondral bone is replaced with granular tissue, edema within the marrow occurs, and sclerosis on the margins takes place, revascularization and reconstruction can still be achieved [15]. Complete revascularization of the talus after surgery or injury takes 6 months to 3 years [16]. In stage III, due to a continuous lack of oxygen supply, the necrotic area surpasses the level at which bone congruity can be maintained which causes the subchondral bone to collapse, resulting in joint incongruity. At this stage, surgical treatment is necessary in most cases. In stage IV, the collapse of the talus continues as the joint space narrows and subchondral cysts and the destruction of the tibial joint space occur. As most of the blood supply to the talus comes from the posteromedial side through the sinus tarsi, partial AVN mostly involves the lateral talus [17].

## 3. Signs and Symptoms

### 3.1. Talar Neck Fracture

Most AVN develops following a talar neck fracture. Urgent management and treatment are crucial for recovery and avoidance of AVN. The degree of risk of AVN development based on talar fracture type is summarized in Table 1, Table 2 and Table 3. Only a Hawkins type I fracture can be treated conservatively—with 8 weeks of non-weight bearing and rigid immobilization. Hawkins type II–IV fractures of the talus are treated with open reduction and internal fixation (ORIF) using screws or, more precisely, screw plate fixation, when the patient’s condition precludes surgery [18,19,20]. Talus fractures in the pediatric population can be as devastating as in the adult population and may eventually require arthrodesis. As in adults, the greater the age and severity of the fracture, the greater the rate of AVN; however, non-union was observed only in patients older than 12 years [21].

### 3.2. Diagnosis

The first signs of avascular necrosis of the talus can be observed 6–8 weeks post fracture, with progressive loss of range of motion and talar impingement due to the collapse of the talar dome [23].

The first line of diagnosis consists of X-ray imaging, which should be performed 6–8 weeks post fracture. At this point, the first red flag is the absence of the Hawkins sign. The Hawkins sign is a subchondral radiolucent band in the dome of the talus. This results from an increase in bone reabsorption relative to bone formation, and it manifests with active hyperemia of the bone [24]. It appears mostly between 6 and 12 weeks post fracture. The absence of Hawkins sign is, in 58% of cases, indicative of AVN; however, its presence can in almost 100% of cases exclude AVN, although its role here is being questioned [25]. The disease, like in other AVNs, is classified using the Ficat and Arlet scale (Table 4) [26].

In the case of persisting absence of the Hawkins sign and the presence of talar AVN symptoms after 12 weeks, MRI should be performed [27].

### 3.3. Role of MR

In high-risk patients with clinical suspicion of AVN and negative radiographs, or post-traumatic patients with a negative Hawkins sign, MRI should be performed. Any high-signal-intensity line in the talar dome fat-saturated in T2-weighted MR or a serpiginous low-signal-intensity line in the talar dome T1-weighted MR combined with clinical examination may indicate AVN. In the case of post-traumatic AVN treated surgically, the role of MRI is limited due to the presence of stabilizing hardware. In the case of non-operative treatment of a talar fracture, MRI should not be performed earlier than 3 weeks post talar neck fracture [28]. It is important not to miss the early stages of atraumatic AVN, as in the case of potential progression to stage III or IV, there is no chance for macrostructural restoration, and the patient may require immediate surgical treatment. AVN should not be misdiagnosed for processes including infection, stress fractures, OCD, neoplasm, or primary osteoarthritis [15].

### 3.4. Treatment Options

As the authors consider the treatment algorithm developed by Dhillon et al. as viable, the treatment section will consist of a summary of surgical techniques used at each stage [29]. According to Dhillon et al., it is first necessary to determine the advancement of the disease (stage I–IV) and its cause (post-traumatic/non-traumatic). In general, in stages I–II, conservative treatment (non-weight bearing + ESWT, hyperbaric oxygen treatment) or joint-sparing procedures, including core decompression and bone grafting, should be applied. In the case of disease progression or diagnosis in stages III and IV (post-traumatic/non-traumatic), either alloplasty (total talar replacement/combined total ankle replacement) or salvage procedures, including Blair tibiotalar (TT) fusion, ankle fusion, hindfoot arthrodesis, and tibiotalocalcaneal (TTC) fusion, should be performed.

## 4. Conservative Treatment

### Non-Weight Bearing

Non-weight bearing is the first line of treatment mostly for patients with post-traumatic talar AVN in stages I and II on the Ficat and Arlet scale. Non-weight bearing allows the subchondral bone to revascularize and restores proper bone structure. Thus, non-weight bearing in stages III and IV, where revascularization is no longer possible, is not appropriate [28]. Five studies regarding non-weight bearing, including one RCT, have been conducted; most authors recommended that the period of weight bearing prohibition should not exceed 3–6 months [30,31,32,33]. As Dhillon et al. suggest, as soon as bone union is visible in exams, weight bearing should be increased within the limits of pain. The effectiveness of solely non-weight bearing ranges from 29 to 55%; however, as proven in the RCT by Zhai, this might be enhanced to up to 77% using liquid–electric ESWT.

Non-weight bearing may be similarly augmented with bone marrow aspirate injections, which have been proven to diminish the rate of collapse (progression to stage III) in post-traumatic femoral head AVN. No clinical trials on talar AVN have been conducted.

Hyperbaric oxygen treatment in the early postoperative period has been described as effective in a three-year perspective in a dual case report of patients with Hawkins type III fractures [34].

## 5. Joint-Sparing Treatment

### 5.1. Core Decompression

Core decompression is a surgical option mostly reserved for non-traumatic talus AVN in stages I, II, and, in some cases, III. Core decompression using small diameter K-wires decreases intraosseous pressure, enabling revascularization in the necrotic area. The treatment is often compounded by non-weight bearing and surgery being performed with a posterolateral approach. According to published research, the effectiveness of core drilling in non-weight-bearing conditions—defined by non-progression of the disease and good to excellent daily functioning—ranges from 82% to 91% for stages I and II [35,36,37,38]. Relief of symptoms is usually seen in 3–6 months. Core drilling can be repeated in cases of inefficient revascularization. In clinical practice, based on reported cases, core drilling should be mixed with non-weight bearing. The efficiency of core drilling in post-traumatic cases has not been described.

### 5.2. Bone Grafting

Vascularized and non-vascularized bone grafts are suitable options, both for post-traumatic and idiopathic cases in stages II–III, with satisfactory results ranging from 83.3 to 90% of cases [39,40,41,42,43]. Bone grafting can be performed along with core decompression in stages II and III or joint arthrodesis in stages III and IV. Interestingly, serious bone deficits may be successfully treated with femoral head allograft, with significant improvement of physical symptoms, namely, FAAM (foot and ankle ability measures) activities of daily living (ADL) and sport, AOS (ankle osteoarthritis scale) pain and disability, and the VAS (visual analogue scale) score, as presented by Coetzee et al. [44]. However, it is important to note that this study was primarily concerned with large osteochondral defects and may not directly apply to avascular necrosis.

## 6. Joint-Sacrificing Surgical Procedures

Salvage procedures are indicated for stages III and IV, when all previous measures have failed. The goal of salvage procedures is to achieve a painless and well-aligned plantigrade foot (0–5° of hindfoot valgus and 5–10° of external rotation). The main disadvantages of the fusion include a loss of motion in the ankle joint, arthritis developing in the surrounding joints, and abnormal gait resulting in adjacent joint arthritis.

All the procedures involve partial or total talectomy in order to maintain the limb length.

### Blair Tibiotalar (TT) Fusion, Arthroscopic Ankle Fusion, Hindfoot Arthrodesis, and Tibiotalocalcaneal (TTC) Fusion

Blair tibiotalar fusion involves the excision of the talar body, preservation of the talar neck, optional introduction of bone grafts, and internal fixation with various fixators (Figure 3). The advantages of Blair fusion include the preservation of subtalar motion and limb length, better cosmesis, and shoe wearing. It can be performed using a hip compression screw, modified bone staple, anterior plate fixation (Van Bergey et al.), or anterior compression plate (Ross et al.) [45,46]. Trials performing fusions with bone augments have reported greater efficiency and union rates of vascularized compared to non-vascularized grafts [47].

Other forms of ankle arthrodesis encompass similar advantages and disadvantages to Blair fusion. As described by each of the authors, the success rate is high, but the risks of the procedures include graft collapse, donor site morbidity, infection, and non-union. Various hindfoot arthrodeses were performed for symptomatic osteonecrosis of the talus, with the authors achieving union and satisfactory results in 82% of fusions, observed radiologically after 7 months [48]. Another option includes arthroscopic ankle fusion for AVN of the talus. As stated by Kendal et al., in a cohort of sixteen patients with fusion occurring each time [26], the rate of success of TTC with the use of retrograde intramedullary nail or retrograde compression intramedullary nail ranged from 85 to 100%, which can be augmented with bone allograft [49,50,51,52]. In cases of large segmental deficit, as an interesting alternative to bone graft, Cohen and Kazak described the use of a porous tantalum spacer, an autogenic morselized fibular bone graft, and 30 mL of bone marrow aspirate in conjunction with a retrograde tibiocalcaneal nail, achieving satisfactory results [53]. The outcomes of talar arthrodesis with the use of circular external fixators were unsatisfactory [54].

It is worth noting that all these studies provided predominantly level 4 evidence, the number of patients was relatively low, and there was significant heterogeneity among the population reported.

## 7. Replacement Surgery—A Modern Alternative for Stage III and IV Talar AVN

Partial talar replacement, TTR, and combined TAR are procedures dedicated to “post collapse” stages III and IV as an alternative to joint-sacrificing procedures. Up-to-date studies are included in Table 5, Table 6 and Table 7.

### 7.1. Partial and Total Talar Replacement

The main concerns regarding ankle desis include (i) delayed union, (ii) a shortening of the limb, (iii) a lack of mobility, and (iv) the stiffening of adjacent joints. To counter these problems, talar prostheses were invented and at first widely used by Harnroongroj [58]. Figure 4 presents a scheme of three generations of talar protheses. The first-generation TTR was a partial talar prothesis mounted on a peg and cemented. Unlike the ankle arthrodesis, the prostheses allowed movement in the ankle joint as well as maintenance of limb length. Talar prostheses were perfected in Japan. The second generation did not include a peg; it was mounted solely on cement. Despite good clinical results, the problem with the prothesis was settling it deep into the talus bone [56]. Hence, the third-generation implant was created, which, in contrast to previous ones, completely replaced the talus bone. Current talar prostheses are manufactured based on a 3D CT image of the contralateral talus, performed in 2 mm intervals. Protheses are made of stainless steel, aluminum–ceramic, cobalt–chrome, and vitallium alloys, and are now increasingly covered with titanium [59]. The advantages of the talus prosthesis, as opposed to the arthrodesis, are (i) a brief period of weight-bearing restriction, (ii) quick pain relief, and (iii) the preservation of the limb length. Potential disadvantages of a total talar prosthesis include (i) implant dislocation/fracture, (ii) peri-talar instability, (iii) implant migration, (iv) degenerative changes in the surrounding tarsal joints, and (V) the cost of the procedure relative to the desis.

Moreover, an important disadvantage is anterior instability in cases of ATFL and deltoid ligament dissection. The effects of TTR are assessed based on ankle mobility, the AOFAS scale (90 points, the more the better: 40 for pain, 50 for activity limitations, walking distance, walking surface, gait abnormality, sagittal motion, hindfoot motion, ankle hindfoot stability), the SF 36 scale (physical functioning), the JSSF scale, and the VAS pain scale (10 to 1). Drawing clear conclusions on talar prostheses is difficult due to (i) varying observation periods, (ii) varying materials from which the prostheses are made, (iii) varying surgical approaches, and (iv) varying postoperative protocols included in trials. Protheses made of stainless steel, implanted in the years 1974 to 1990, used a medial approach. The results were described by Harnroongroj as excellent for three patients after 5 years, one experienced failure within 5 years, three patients were satisfactory after years 6 to 9, and almost all were satisfactory (8/9) after 11 years. Complications arose from the fact that the stem sunk into the neck of the talus bone, as the circumference of the lower part of the prosthesis was too small, resulting in erosion and settling on the posterior facade of the calcaneus.

In 30 years of follow-up, 28/33 prostheses were still inside the talus, and the AOFAS score did not deteriorate compared to the period immediately after surgery [60]. Late complications included (i) prothesis failure due to size mismatch, (ii) neoplasm recurrence, and (iii) infection. 

**Table 6 medicina-60-01692-t006:** Total talar replacement—included studies.

Study	Study Type	Patient Count	Implant Type	Follow-Up	VAS	ROM	SF 36	AOFAS	Complications
Ando et al., 2016 [61]	Case report	1	Custom-made alumina ceramic	2 years	-	Plantarflexion 40° Dorsiflexion 20°	-	90 (45)	-
Harnroongroj et al., 1997, 2014 [58,60]	Case series	23 (33)	Stainless steel talar body	10–20 y (8) 20–30 y (11) 30–36 y (9)	-	10–20 y Plantar flexion 22–38° Dorsiflexion 0–10° 20–30 y Plantarflexion 5–32° Dorsiflexion 0–5° 30–36 y Plantarflexion 10–30° Dorsiflexion 0°	-	10–20 y—78 20–30 y—76 30–36 y—76	Erosion, collapse of ON head or neck, hypertrophic ossification
Scott et al., 2020 [62]	Case series	15	Cobalt–chromium total talus 3D-printed	1 year	3.6	Increased postoperative ROM: ankle 5° subtalar 6°	-	-	Wound dehiscence (n = 1)
Morita et al., 2022 [63]	Case series	19	Customized alumina-ceramic total talar prosthesis	10 years	-	Dorsiflexion 7° Plantarflexion 45° Total ROM 45°	-	-	Postoperative numbness at the medial side of the ankle (n = 1)
Taniguchi et al., 2015 [64]	Case series	55	Alumina-ceramic total talar prosthesis	4.4 years	-	Dorsiflexion 5.4° ± 4.9° (range, 0° to 17°) Plantarflexion 32.0° ± 8.9° (range, 17° to 55°)	-	-	-
Tonogai et al., 2017 [65]	Case report	2	Custom-made alumina-ceramic total talar prosthesis	1.5 years	0	Dorsiflexion 10° and 10° Plantarflexion 30° and 40°	-	-	-
Tracey et al., 2018 [66]	Case series	14	Nickel-plated cobalt chrome total talar prosthesis	21 weeks	-	-	-	-	-
Kadakia et al., 2020 [67]	Case series	27	Cobalt chromium ± titanium nitride coating	22.2 months	3.9	Total ankle ROM 49.4° Dorsiflexion 6.9° Plantarflexion 42.5°	-	-	- Superficial peroneal nerve neuroma - Below knee amputation due to persistent pain - Significant distal tibia AVN without collapse
Angthong 2014 [68]	Case series	5 (1 AVN)	Stainless steel	17.8 months	-	Substantially satisfactory	83.38	-	-
Abramson et al., 2021 [69]	Case series	8 (4 AVN)	Titanium with ceramic surface treatment for a titanium oxide surface layer	23.1 months	-	-	83.25	79.25	-
Ouchi et al., 2023 [70]	Case series	10	Custom-made ceramic total talar prosthesis	5 years	1.8	Dorsiflexion 11 ± 6° Plantarflexion 38 ± 4°	-	-	-
Luo et al., 2022 [71]	Case series	4	Custom-made vitallium prosthesis	29–78 months	3, 0, 6, 1	Dorsiflexion 30° Plantarflexion 20°	-	88.5	Tingling and numbness on the skin of the ankle
Madi et al., 2022 [72]	Case series	2	3D-printed cobalt-chromium talar prosthesis	3 years and 1 year	6 and 8	Dorsiflexion 10° and 5° Plantarflexion 30° and 30° Total ankle ROM 40° and 35°	-	-	Constant pain below knee amputation 3 years post TTR

The most extensive study on second-generation prostheses was conducted by Taniguchi [56]. A total of 22 patients underwent TTR: 8 of them had a first-generation prosthesis, 14 had a second-generation prosthesis, and the average follow-up time was 8 years (98 months). Despite acceptable results, some patients required revision due to implant loosening and migration. In summary, despite satisfactory results, it was suggested that partial talar implants (first and second generations) should not be used in the future, only third-generation prostheses.

Third-generation protheses may be categorized as constrained (with fixation into the calcaneus or with ligamentous reconstruction) or unconstrained (growth into the prothesis). These are composed of ceramic–aluminum, cobalt–chromium, stainless steel, vitallium, and titanium.

The most extensive study covering third-generation aluminum prostheses, with a mean follow-up time of 4.4 years, originated from A. Taniguchi’s team [64]. Significant improvement in pain and improvement in range of motion were reported. Early outcomes of TTR were favorable, but the current production method is expensive and time-consuming.

Similarly, extensive studies have been conducted by D. J. Scott (2 years of observation), S. Morita (13 years of observation), and Kadakia, who investigated cobalt–chrome and alumina-ceramic prostheses [62,63,68]. The numbers of patients were 15, 18, and 27, respectively. The reoperation rate was lower than that with traditional arthrodesis [61]. Osteophyte development and sclerotic changes were observed in adjacent joints but had no effect on the AOFAS and JSSF, which improved. Consecutively, Abramson and colleagues performed TTR on seven patients [69]. Despite satisfactory results, the patients demonstrated a mildly (and, for one of them, a moderately) abnormal gait. The patient with the longest duration of follow-up showed radiological changes of tibial wear, although they remained symptom-free. W. Luo et al. implanted a titanium vitallium prosthesis with good results [73]. Numerous case reports in the literature have presented satisfactory results, although as opposed to theory, the mobility of the ankle joint was often incomplete.

The long-term results of first-generation prostheses were satisfactory in terms of reducing pain and maintaining limb length. The complication rate was acceptable. There are reasons to believe that the current third-generation prostheses will provide a great alternative to arthrodesis. The main limitation now is the cost of the procedure and the lack of general availability of implants.

A different procedure was performed by E. Ramhadany, which involved combining TCN (tibiocalcaneal nail) with a special 3D titanium cage and nail attachment [57], with a follow-up period of 36 months. AOFAS increased from 6 points preoperatively to 28/37 points. After one year, union was achieved radiologically.

### 7.2. Total Ankle Replacement and Combined Total Talar Replacement

In combined total talar replacement, the talar components consist of third-generation total talar prothesis, an appropriately sized polyethylene mobile bearing, and a tibial component that comes from a TAR set. In late 1980s, S. Newton III, in one of the earliest studies, described AVN of the talus as a contraindication for TAR [74]. However, since the implementation of total talar prothesis (not only its partial counterpart), there is no longer a risk of talar component subsidence.

**Table 7 medicina-60-01692-t007:** Combined total ankle replacement—summary of current studies.

Study	Study Type	Patient Count	Implant Type	Follow-Up	VAS	ROM	SF 36	AOFAS	Complications
Devalia et al., 2015 [73]	Case series	7	First stage—subtalar fusion, Second stage—mobility total ankle replacement (DePuy International)	36 months	-	-	Improved from pre-op to 3 years post-op for 6/8 domains	77.5	-
West and Rush 2021 [75]	Case series	3	Cobalt chromium	13.7 months	-	-	-	-	-
Lee et al., 2008 [76]	Case series	2	Three-component mobile-bearing HINTEGRA prosthesis	30 and 24 months	-	Dorsiflexion 5° Plantarflexion 30°	-	91 and 85	Anterior translation of the talar component—no loosening
Madi et al., 2023 [77]	Case series	9	Cobalt–chrome 3D-printed talus. TAR—Cadence, Inbone, Salto Talaris	19.4 months	1.8	Dorsiflexion 11.6° Plantarflexion 33.3°	-	-	Varus ankle deformity (n = 1)
Chinzei et al., 2021 [78]	Case report	1	Alumina ceramic talar prosthesis and TNK tibial component (Kyocera)	4 years	-	Dorsiflexion 5° Plantarflexion 15° Inversion 5° Eversion 0°	-	88	-
Buchholz et al., 1977 [79]	Case report	1 (bilateral)	St. Georg–Buchholz prosthesis	18 months	-	Almost complete motion in both ankle joints	-	-	-
Kurokawa et al., 2019 [80]	Case series	10	TNK ankle (Kyocera, Kyoto, Japan) and an alumina-ceramic artificial talus (Kyocera)	58 months	-	-	-	-	-
Kanzaki et al., 2019 [81]	Case series	22 (18 AVN)	TNK tibial component (Kyocera) and custom-made total talar prosthesis (Kyocera)	-	-	Dorsiflexion 14.4° Plantarflexion 32° Total ROM 46.5°	-	-	- Intraoperative medial malleolus fracture (n = 1) - Postoperative medial malleolus fracture (n = 2) - Delayed wound healing (n = 3)
Yamamoto et al., 2022 [82]	Case series	26	TNK ankle and alumina-ceramic talus (Kyocera)	46 months	-	Dorsiflexion 15.5° Plantarflexion 34.2°	-	-	- Superficial infection (n = 1) - Periprosthetic fracture (n = 3)

The use of the aforementioned hybrid is appropriate for (i) AVN of the talus post TAR procedure and (ii) osteoarthritis of the ankle joint with collapse and AVN of talar body. The advantage of combined TAR over sole TTR is the avoidance of cartilage wear of the distal tibial articular surface.

This concept was proven by Kurokawa on 22 patients who compared performance post standard TAA with combined TAR [80]. The patients who underwent combined TAR presented better JSSF (Japanese Society for Surgery of the Foot) and SAFE-Q (Self-administered foot evaluation questionnaire) scale scores than standard TAA. Nonetheless, only one patient suffered from AVN, not OA. Similarly, satisfactory clinical results using combined TAR were presented by Kanzaki, who described combined TAR in 22 patients, reporting mean ROM improvements from 4.0 to 14.4 degrees in dorsiflexion and from 23.8 to 32.0 degrees in plantarflexion; the JSSF scale score improved by 40 points [81]. Complications included prolonged wound healing and medial malleolus fracture. Confirming results were presented in case reports by T. A. West, K. B. Lee, and N. Chinzen [75,76,78].

By contrast, in the most extensive study (T. Yamamoto) that compared combined TAR and standard TAR (69 patients), no difference was found in terms of JSSF scale and SAFE-Q between the two groups [81]. A higher revision rate in traditional TAR was reported.

Another approach to TTR was proposed by Devalia et al. [73]. Firstly, hindfoot arthrodesis was performed. Then, after revascularization had taken place, total ankle prothesis was introduced. The procedure was performed in seven patients, and statistically significant improvement in WOMAC, SF 36, and AOFAS was confirmed. Complications in two patients included implant settlement at 2-year follow-up, although this had no impact on patients’ clinical outcomes.

The last method by S. J. Madi encompasses combined TAR with hindfoot arthrodesis [77]. In case series encompassing nine patients and 2 year follow-up, there were statistical improvements in the VAS score, ankle plantarflexion, talocalcaneal height, and tibiotalar alignment. Patients achieved a successful union of their subtalar and talonavicular joint arthrodesis.

## 8. Discussion

The increasing availability of MRI is likely to lead to a higher incidence of early and idiopathic talar avascular necrosis (AVN) detection. MRI’s sensitivity allows for the identification of AVN at stages when intervention can prevent progression to more severe and debilitating forms of the disease. This shift towards earlier diagnosis could significantly improve patient outcomes by enabling timely treatment.

As technology advances, the use of custom-made total talar prostheses is expected to become more prominent in managing AVN. These prostheses, tailored to the specific anatomical needs of each patient, offer enhanced fit and function, improving the overall success rates of talar replacements.

Additionally, the role of total ankle arthroplasty (TAA) is anticipated to grow, particularly as a treatment option for patients with advanced AVN. TAA offers a viable alternative to traditional joint-sacrificing procedures, preserving joint mobility and improving the quality of life for patients with severe talar damage. This evolution in surgical options reflects a broader trend toward more personalized and functional solutions in orthopedic care.

## 9. Conclusions

Talar AVN, though uncommon, is a challenging clinical condition to treat. All patients, even pediatric patients suffering from talar neck fracture, should be observed for signs of talar AVN for a period of at least 3 months. It remains crucial to watch for the absence of the Hawkins sign and, in such cases, perform MRI. Similarly, non-traumatic patients suffering from systemic lupus erythematosus (SLE) or type 2 diabetes (DM II), receiving steroid therapy, or reporting chronic ankle pain should be screened for AVN.

Due to the careful screening of all susceptible patients with increasing availability of MRI and an aging population, the prevalence of talar avascular necrosis and its detection, especially in the preliminary stages, will increase. Thus, bearing in mind Dhillon’s algorithm, it remains mandatory to rely on non-weight bearing, ESWT therapy, bone marrow injections, bone grafting, and core drilling adjusted to each patient in stages I and II, as this can translate to greater incidence of successful bone union, shorter periods of patient disability, and lower treatment costs.

## 10. Future Directions

The role of total talar replacement will gain importance in time, as will the availability of third-generation talar prostheses, offering a short period of prohibition of weight-bearing, quick pain relief, partial preservation of joint motion, and preservation of the length of the limb. The 30-year observation of first-generation protheses has proven that TTR is a safe and pain-relieving procedure. The third-generation, full-body protheses may present even better results. More prospective studies are required to confirm the safety and benefits of TTR. The assumed advantage of combined TAR over standard TAR or TTR is still debatable and requires further examination. Until then, ankle fusion, hindfoot arthrodesis, and tibiotalocalcaneal (TTC) fusion remain cheaper and more reliable alternatives in post-collapse AVN stages III and IV.

## Figures and Tables

**Figure 1 medicina-60-01692-f001:**
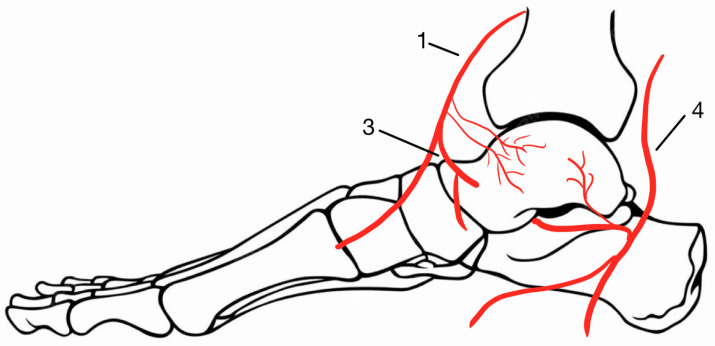
Blood supply of the talus: 1—anterior tibial artery, 3—medial talar artery, 4—posterior tibial artery.

**Figure 2 medicina-60-01692-f002:**
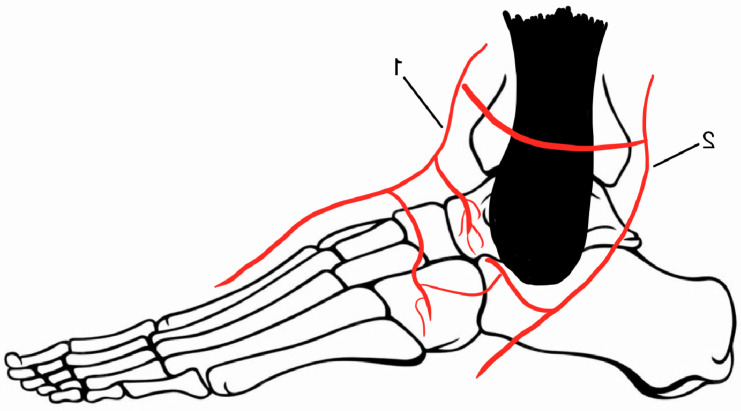
Blood supply of the talus: 1—anterior tibial artery, 2—perforating peroneal.

**Figure 3 medicina-60-01692-f003:**
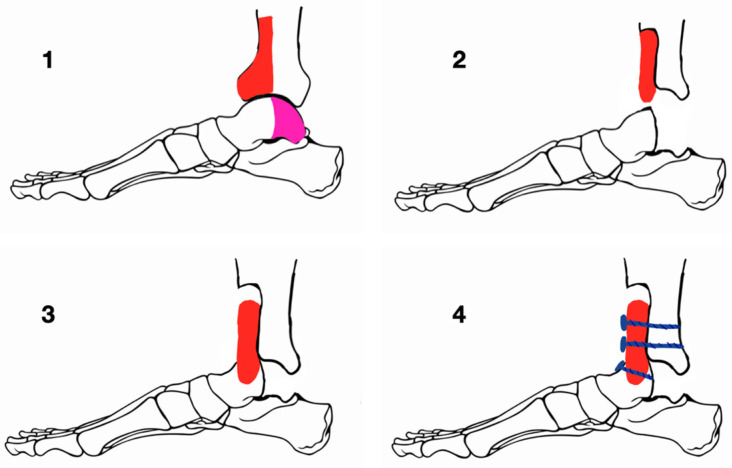
Blair fusion. 1–4 consecutive stages of the surgical procedure; pink zone—the resected talus body; blue lines—cortical screws.

**Figure 4 medicina-60-01692-f004:**
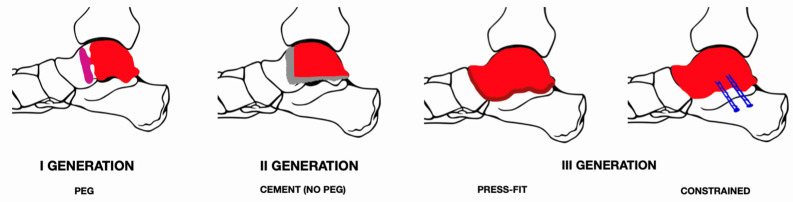
Total talar prothesis: scheme of the three generations. Red color—talar prothesis, grey—cement, blue—screws.

**Table 1 medicina-60-01692-t001:** Classification of talus fractures regarding AVN risk (Marti–Weber classification).

	Marti–Weber Classification	Circulation Status	AVN Risk
Type I	Distal talar neck and head fractures. Peripheral fractures and osteochondral flakes.	Intact	No necrosis
Type II	Nondisplaced talar neck and body fractures	Mainly intact	Seldom
Type III	Displaced talar neck and body fractures	Intraosseous circulation injured Supplementary circulation intact	Often
Type IV	Proximal talar neck fracture, luxation of corpus tali, comminuted fractures	Both injured	Almost always

**Table 2 medicina-60-01692-t002:** Classification of talus fractures regarding AVN risk (Hawkins classification).

	Hawkins Classification	AVN Risk
Type I	Nondisplaced neck fracture	0–13%
Type II	Subtalar dislocation	20–50%
Type III	Subtalar and tibiotalar dislocation	20–100%
Type IV	Subtalar, tibiotalar, and talonavicular dislocation	70–100%

**Table 3 medicina-60-01692-t003:** Classification of talus fractures regarding osteonecrosis (Hawkins classification).

	Modified Hawkins Classification [22]	Osteonecrosis	Osteonecrosis with Collapse
Type I	Nondisplaced neck fracture	0	0
Type IIA	Subluxated subtalar joint	0	0
Type IIB	Dislocated subtalar joint	25%	19%
Type III	Dislocated subtalar and tibiotalar joint	41%	19%
Type IV	Subtalar, tibiotalar, and talonavicular dislocation	33%	33%

**Table 4 medicina-60-01692-t004:** Ficat scale for Talar Avascular Necrosis.

Stage	X-ray Findings
I (Preradiographic)	Normal
II (Precollapse)	Bone remodelling, subchondral cysts
III (Post collapse)	Crescent sign
IV (Arthritis)	Talar bone deformation, joint space narrowing

**Table 5 medicina-60-01692-t005:** Partial talar replacement—included studies.

Study	Study Type	Patient Count	Implant Type	Follow-Up	VAS	ROM	SF 36	AOFAS	Complications
Cui et al., 2023 [55]	Case report	1	3D titanium alloy partial talar prosthesis	2 years	1	Improvement in active dorsiflexion Plantarflexion 45°	-	93	-
Taniguchi et al., 2012 [56]	Case series	22	8 alumina-ceramic talar body prostheses with peg 14 alumina-ceramic talar body prostheses without peg	8.2 years	-	-	-	1st generation 80 2nd generation 81.1	- Loosening around the peg and necrosis of the talar head in all patients with peg prostheses - Motion between native talus and talar body prosthesis in half of second-generation prostheses
Ramhamadany et al., 2021 [57]	Case series	3	Keystone-shaped, custom-made, 3D-printed titanium truss implant	32 months	-	-	-	Modified score 37 (max 60 points) at 24 months FU	- Breakage of the lag screw perforating the talonavicular joint

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
