# Peer review of "Avascular Necrosis of the Talus: Diagnosis, Treatment, and Modern Reconstructive Options"

_medicina, 2024, doi:10.3390/medicina60101692_

Round 1

Reviewer 1 Report

Comments and Suggestions for Authors

change title:   Avascular necrosis of the talus-diagnosis, treatment and modern reconstructive options

abstract: stages 3-4 require more "advanced" procedures

sentence starting with development of 3D printing...  needs to be reworded

throughout paper the word "deses"  should be replaced with arthrodesis

page 3 :  necrosis of the bone, which "can lead to" collapse ...

Table 1 would be easier to read if it was dividing into 3 parts corresonding to each classification.  also should cite Vallier and divide the Hawkins 2 into A and B(A is displaced neck fracture without dislocation of ST joint and B is neck fracture with dislocation of ST joint)

second to last paragraph: risk factors should include whether or not there is subtalar dislocation

page 4: first paragraph:  change to Recently, iatrogenic talar avascular necrosis has been described following subchondroplasty of the talus with calcium phosphate injections.   OCD should be spelled out.

paragraph 3:  perfiferic  should be proliferative

odema should be edema within

... bone congruity can be maintained which...

...surgical treatment is necessary.In stage IV the collapse of the talus continues as subchondral cyst form and there is destruction of the tibial articular surface. 

... change mostly to most commonly

3.  Signs and symptoms

Most AVN develops following a talar neck fracture.  Urgent management ...

Delete external fixation.  end sentence after screw plate fixation

Diagnosis:

most displaced talar neck fractures are treated with 12 weeks of non weight bearing following fixation.  Recommend deleting first paragraph.

the role of the Hawkins sign is being questioned as it does not really change treatment in the early stages of recovery.  for this paragraph, just describe what the Hawkins sign consists of 

I disagree that MRI is indicated.  with hardware in place the MRI is not helpful and really does not change management, so I do not think it should be recommended.

Under role of MRI, I do not believe there is any widely accepted treatment for signs of AVN in late stage 2 or early stage 3.   many patients have signs of AVN and it does not progress.  we do not know what the progression is from stage 2 to 4.   it is not the same as the hip which is where the Ficat classification was developed.  you cannot extrapolate those numbers to the talus as the progression is not the same.   Recommend deleting the whole section on MRI.

TTR and TAR  (this should be new section)  Delete this paragraph and start new section titled "Treatment options"

There is no proof that non weight bearing decreases incidence of post traumatic AVN.  A short period 2-3 weeks of nwb may be indicated in nontraumatic AVN to manage associated pain.  In cases of traumatic AVN follwing fracture, generally patients are kept non weight bearing for 12 weeks and then advanced according to their pain.  in modern times , it is not realistic to keep a patient non weight bearing for more than 12 weeks.

just list procedures like bone marrow aspirate injection, core decompression, hyperbaric oxygen, bone stimulation as treatments that have been tried , but only level IV evidence is reported.  this cannot be recommended based on literature available.

Core drilling : change to Core Decompression

we do not know the number of stage 1 and 2 that progress to stage 3, so the numbers reported on results of core decompression do not mean much as many of the people with stage 1 and 2 do not progress to collapse anyway.

Bone Grafting:  Stadia?

the results of the Coetzee study are more related to large osteochondral defects, not necessarily Avascular necrosis.

spell out FAAM,  AOS and VAS

Salvage procedures are indicated

abnormal gait resulting in adjacent joint arthritis

joint instability is not a problem with hindfoot fusion

Delete "Blair's tibiotalar fusion, arthroscopic ankle fusion,hindfoot arthrodesis, "

it would be helpful to have diagram of Blair fusion.  I do not think it is used much now and is of historical note only.  discussion of hip compression screw, etc is of historical interest only and should not be recommended

The rate of success should be noted.  I would argue that the rate of success of an isolated ankle fusion for AVN is not high, that is why there are so many other options being explored.  not sure where they came up with 7 months.

The problem with all these studies is that the numbers are very low and there is significant variability in the patients reported.  The authors should say that these are predominantly level 4 studies in highly variable patient populations.

to really describe the management, they should separate procedures that preserve the talus and those that excise the talus and replace it with either allograft or a prosthetic replacement.  again these are very small series or case reports.

In tables 3 and 4, they should note the year the study was published.

under TTR:  it would be helpful to see pictures of the different prostheses.  Again, many of the studies cited are of historical interest only as these prostheses are no longer in use.  the japanese studies have the longest follow up and these should have pictures.

should expand the discussion and divide paragraphs into first generation and the years used followed by second generation and years used, etc

I do not think anyone has proven that anterior instability is a problem.  most people are stiff, thus not unstable

the first generation of prosthesis they describe is a partial replacement.  need to separate total talus from partial talus discussions

paragraph starting with 3rd generation   ... unconstrained means there is no adjacent fusion or ligament attachment.  do not use term "press fit" this implies in growth into the prosthesis which would make it constrained

delete paragraph starting with "Alternative"

"isolated clove"  i do not know what this is

need to spell out JSSF and SAFE/Q scale;   not familiar with these scores

custom made and personalized are the same, delete one of them

conclusions: Talar AVN is a challenging clinical condition.

Talar neck fractures should be observed for at least a year, probably 2 years.

in Vallier's study, AVN presented at an average of 11 months post injury.

I would recommend MRI only in non traumatic cases.

None of the options for stages 1 and 2 are proven in clinical studies,  cannot recommend them, just list them as options that have been tried.

Comments on the Quality of English Language

comments are in the section for the authors and editors regarding the english.

Author Response

Dear Reviewers,

Thank you for your valuable feedback on our manuscript. We have carefully considered all of your comments, with the exception of one pertaining to modifications in the discussion section.

We would greatly appreciate any further guidance you could provide on how to effectively address this point.

Thank you once again for your support.

Best regards,

Reviewer 2 Report

Comments and Suggestions for Authors

Dear Authors,

Overall a high-level review of Talar avascular necrosis and the mechanism behind it and the treatment. As this was a high-level review with great figures and tables. There is minimal suggestions by this reviewer outside of a few minor changes.

1.The title is long and will not have great SEO. I suggest a shorted title: Talar avascular necrosis: diagnoses, treatment and total replacement.

2. Keep the tables consistent in formatting as some the boxes have outlines and others do not.

Author Response

(The authors gave the same response as above.)
